# Cognitive Characteristics of an Innovation Team and Collaborative Innovation Performance: The Mediating Role of Cooperative Behavior and the Moderating Role of Team Innovation Efficacy

**Mi Zou [1,2], Peng Liu [1,*], Xuan Wu [1], Wei Zhou [1], Yuan Jin [1] and Meiqi Xu [1]**

[1] School of Management, Zhengzhou University, Zhengzhou 450001, China; zouminyist@163.com (M.Z.); wx548246426@163.com (X.W.); m18695906827@163.com (W.Z.); 15738967075@163.com (Y.J.); 17737531228@163.com (M.X.)

[2] School of Teacher Education, Nanyang Institute of Technology, Nanyang 473004, China

\* Correspondence: liupeng@zzu.edu.cn

**Abstract:** Based on the theory of social cognition, this paper discusses the cognitive characteristics of an innovation team, the influence mechanism of cooperative behavior on collaborative innovation performance, and the moderating effect of team innovation efficacy in an enterprise collaborative innovation network. The hypothesis has been verified on the basis of 288 valid questionnaires. The results show the following: in the process of collaborative innovation, different dimensions of innovation team cognitive characteristics, namely collaborative innovation experience, the internal innovation environment of the enterprise, and emotional experience, have a significant positive impact on the collaborative innovation performance; and cooperative behavior plays a partial mediating role in the cognitive characteristics of the innovation team and collaborative innovation performance. In addition, the team's innovation efficacy positively moderates the relationship between cooperative behavior and collaborative innovation performance. The results of this study not only expand the application of the social cognitive theory to the scope of collaborative innovation activities, but also have a certain reference significance to effectively mobilize the innovation initiative of the enterprise innovation team and improve the performance of collaborative innovation.

**Keywords:** innovation team; cognitive characteristics; cooperative behavior; collaborative innovation performance; team innovation efficacy





## 1. Introduction

The uncertainty of technological innovation, coupled with the fragmentation of innovation resources, makes it imperative for enterprises to engage in collaborative innovation with external organizations in order to compensate for their own innovation capabilities and thereby enhance their innovation output. However, there is a general lack of motivation, a weak concept of cooperation, and low efficiency in collaborative innovation activities. Therefore, it is becoming increasingly important to study how to play the mediating role of cooperation behavior to enhance the value creation of collaborative innovation [1].

In this regard, ref. [2] argues that innovation is a process of reconstructing social perceptions, where the internal perceptions of individuals are highly relevant to the exploitation of innovation opportunities. Furthermore, the social uncertainty attitudes, self-confidence, and experience all influence individual thinking and decision making, while innovation teams, as the core of a company's role in the collaborative innovation process, understand and utilize the innovation resources in different ways when carrying out activities related to innovation goals, depending on the level of cognition of the innovation team. This depends on the cognitive level of the innovation team [3], as it determines the approach to innovation that the team takes in its innovation activities, and thus accumulates tacit

knowledge that is attached to the team and difficult to replicate, in order to form stable values that guide the company's innovation activities [4]. Therefore, improving the cognitive level of the innovation team plays a pointer role in promoting the internal development of the company, and it can be argued that the cognitive level of the innovation team is a prerequisite for the collaborative innovation within the company [5,6].

In conjunction with the social cognitive theory, which emphasizes that motivation is the result of a combination of individual factors, such as emotion, experience, and confidence, as well as environmental factors, existing research on collaborative innovation suggests that it is difficult to accurately describe the changes in collaborative innovation performance by simply exploring the one-way relationship between the explicit innovation resource inputs and the collaborative innovation performance [7]. This is because the dynamic, multi-level collaborative innovation activities are influenced by the behavior of many individuals in the process of achieving the innovation goals of the enterprise, which makes it difficult to collaborate effectively and thus limits the output of the collaborative innovation [8]. As research into the relationship between cognition and behavior in innovation management progresses [9], it has been recognized that tacit knowledge or psychological change factors such as experience, environment, and emotional experience are just as important as explicit resources in the innovation process [10].

In the case of collaborative innovation, the innovation output depends not only on the passive behavior of the innovation team, driven by the work they have done, but also on the spontaneous behavior of the innovation team beyond their role as a result of cognitive differences, which makes the relationship between individual emotions, experiences, and behavior increasingly evident, and they may even reinforce the individual behavior [11]. In addition, the output of innovation teams, as key participants in collaborative innovation activities, is also influenced by the team's sense of self-innovation efficacy [12,13]. This is because, as the team's key psychological capital, the team's sense of innovation efficacy is an important motivating and intrinsic support force for its persistence in innovation [14,15]. Therefore, innovation efficacy should be included in the study to explore the level of effort that team members put into accomplishing their innovation goals and their persistence in the face of difficulties in order to demonstrate that innovation efficacy is equally as important in supporting the innovation performance [16–18].

A review of the relevant literature reveals that a large number of studies have identified the consistency of innovation teams' perceptions of their intrinsic innovation environment as an important cognitive characteristic that affects collaborative innovation performance [19,20].

However, there is relatively little research on the correlation between the cognitive characteristics of innovation teams in terms of experience and emotion and collaborative innovation performance. Nevertheless, research in this area has important implications for understanding and improving the effectiveness of collaborative innovation. Therefore, there is a need to further explore the following two aspects:

Firstly, the role of the cognitive characteristics of the innovation team members in supporting collaborative behavior in the collaborative innovation process needs to be examined in depth. Cognitive characteristics include, but are not limited to, experience, knowledge, skills, attitudes, and values. By understanding the differences and commonalities among innovation team members at a cognitive level, the differences in their performance and the effectiveness of collaborative behavior in collaborative innovation can be revealed. For example, experienced team members may play an important role in problem solving and decision making, while emotionally positive team members may be more helpful in promoting team cohesion and creative thinking.

Secondly, there is a need to explore the mechanisms that influence team innovation efficacy in the collaborative innovation process. Team innovation efficacy refers to the team members' confidence and self-belief in the team's ability to innovate and perform creatively. Research has shown that team innovation efficacy is closely related to the innovation performance of a team. Therefore, further research is needed to investigate the

mechanisms of team innovation efficacy on collaborative innovation, including its effects on team members' motivation, collaborative behavior, knowledge sharing, and learning. These studies can help us to better understand the role of team innovation efficacy in collaborative innovation and provide effective support and guidance for the innovation performance of teams.

In summary, although there are relatively few studies on the correlation between the cognitive characteristics of innovation teams and collaborative innovation performance, as well as the influence mechanism of team innovation efficacy in the collaborative innovation process, these issues have important research significance and practical value. Through further in-depth research, we can better understand and promote the effectiveness of collaborative innovation in innovation teams and improve the innovation capability and performance of teams.

Therefore, based on the social cognitive theory, this paper aims to investigate the role of the cognitive characteristics of innovation teams in the process of collaborative innovation and to fill the research gap in this area. Specifically, this study considers the collaborative innovation experience of innovation teams, the internal innovation environment of the firms in which they work, and their emotional experiences during innovation activities as the cognitive characteristics of innovation teams.

First, we will focus on the collaborative innovation experiences of innovation teams, including the experience background, skills, and knowledge base of the team members. These experiences will be considered as the cognitive characteristics of the team, and we will explore how the team members' experiences influence their collaborative behavior in collaborative innovation and how this collaborative behavior has an impact on the team's innovation performance.

Secondly, we will examine the impact of the internal innovation environment in which innovation teams operate on their cognitive characteristics. The intra-firm innovation environment includes factors such as the organizational culture, leadership style, and innovation resources. We will explore how these environmental factors shape the cognitive characteristics of innovation teams and further analyze the impact of this shaping on the teams' collaborative behavior and innovation performance.

In addition, we will focus on the affective experiences of innovation teams during innovation activities, such as the positive emotions of team members, team cohesion, and creative thinking. These affective experiences will be considered as part of the cognitive characteristics of the team, and we will explore how affective experiences influence the collaborative behavior of the team and further investigate the mechanisms by which this influence works on the innovative performance of the team.

By studying these specific areas, we aim to reveal the mechanisms by which the cognitive characteristics of innovation teams play a role in the collaborative innovation process, and to provide scientific references for companies to participate in collaborative innovation for decision making. This study fills a research gap in the relationship between cognition and behavior in innovation teams and provides theoretical and practical guidance for enhancing the effectiveness of collaborative innovation and promoting the development of firms' innovation capabilities [21].

The remainder of this paper is organized as follows: First, Section 2 presents the theoretical foundations and research hypotheses. Then, Section 3 presents the empirical analysis. In Section 4, the research findings and discussion are introduced. Finally, in Section 5, the study conclusions and implications are elaborated.

## 2. Theoretical Background and Hypotheses

### 2.1. Cognitive Characteristics of Innovation Teams and Collaborative Innovation Performance

As mentioned earlier, the social cognitive theory states that cognition is the human processing of information such as attention, perception, memory, representation, and creative thinking [22–27]. It is clear that innovation teams, as the core of enterprises participating in collaborative innovation, are influenced by their tolerant attitude towards

innovation activities, organizational norms, and coordination when cooperating with other organizations [28].

Scholars argue that personal characteristics, such as the cognitive and behavioral tendencies of top management, may also have a significant impact on a firm's ability to engage in exploratory and developmental activities. Relying on insights from cognitive psychology and hypothesizing a relationship between cognitive style and an individual's propensity to explore and exploit, ref. [29] conducted a quantitative study on the relationship between a CEO's personal characteristics and firm-level exploration and exploitation. Furthermore, ref. [30] concluded through their research that executives' goal orientations (the intrinsic motivations that shape what individuals typically seek to accomplish when engaged in challenging tasks) influence their firms' environmental scanning. Specifically, firms whose executives exhibited higher learning goal orientations or higher performance-proven goal orientations were more likely to engage in environmental scanning than those firms whose executives exhibited higher performance-avoidance goal orientations. Ref. [31] examined the antecedents of individual-level innovation performance, the perceptions of entrepreneurial opportunities, and the relationship between the two. The results show that individual self-efficacy, social networks, prior knowledge, and perceptions of the industrial environment of the opportunity all have a positive effect on entrepreneurial opportunity perceptions. Entrepreneurial opportunity perceptions also had a significant effect on individual innovation performance.

An individual's ability is crucial to the outcome of their entrepreneurial activities [9]. Therefore, this paper follows the discussion of the previous studies on the cognitive characteristics of individuals and teams and chooses the innovation experience of innovation teams in their previous collaborative innovation activities (hereafter referred to as innovation experience), the internal innovation environment of the enterprises that they work in (hereafter referred to as internal innovation environment), and the emotional experience during innovation activities (hereafter referred to as emotional experience) as the proxy variables for the cognitive characteristics of innovation teams.

Having extensive experience in team innovation can greatly enhance the team's ability to collaborate and innovate. Firstly, extensive team innovation experience helps team members to become more familiar with the innovation process and methods. Secondly, the innovation experience of team members can enhance the team's synergy ability. Finally, a wealth of experience in team innovation can also help teams to better cope with the challenges and difficulties of the innovation process. Ref. [32] proposed a work environment assessment model for creativity, pointing out that team innovation experience contributes to a creative work environment. Ref. [33] constructed a model of organizational learning, emphasizing that experience has an important role to play in organizational change and innovation. Ref. [34] explored the relationship between team innovation and execution, noting that team innovation experience is important for collaborative team innovation. Ref. [35] argued that firms rely heavily on previous management practices and experiences in the collaborative process, which may result in the transfer of repetitive behaviors to new partnerships. Based on the R&D innovation process perspective, ref. [36] pointed out that the innovation experience has knowledge inheritance attributes that can significantly promote the interaction of explicit and tacit knowledge. In addition, compared with the introduction of new technologies, the collaborative R&D experience performs better in knowledge absorption and utilization, which helps to improve the overall innovation performance of enterprises. Therefore, we propose the following hypothesis:

**Hypothesis 1a (H1a).** *Rich team innovation experience will significantly enhance collaborative innovation performance.*

In organizations, the innovation environment refers to a series of internal factors and conditions that promote innovation, including the organizational culture, leadership style, employee engagement, communication, and knowledge sharing. Firstly, a good internal

innovation environment can stimulate the employees' creative thinking and innovative behavior. Secondly, a good internal innovation environment promotes synergy among team members. Ref. [11] pointed out that the internal innovation environment of an enterprise will affect the individual's behavioral patterns through individual perception, the scope of which has been expanded to include the individual's perception of specific realities such as infrastructure construction, cultural preferences, etc. Enterprises with a strong learning atmosphere will stimulate the participants' innovative thinking and enhance the enterprise's innovation ability, meanwhile, a work environment full of work vitality will reshape the individual's motivation and intellectual resources in a creative way, and thus enhance the individual's innovative vigor.

In summary, a good internal innovation environment has an important impact on collaborative innovation performance. Therefore, we propose the following hypothesis:

**Hypothesis 1b (H1b).** *A good internal innovation environment will significantly enhance collaborative innovation performance.*

Based on past research and theories, we can formulate a hypothesis that positive affective experiences significantly enhance co-innovation performance. Affective experience refers to the emotional states and emotional reactions that individuals have in the work environment, including positive emotions and negative emotions. First, positive affective experiences contribute to the team members' job satisfaction and emotional connectedness. Second, positive affective experiences can promote the development of creative thinking and innovative behaviors. Ref. [37] argued that, in the innovation process, the team members' emotional perception of the leader's management style is key to ensure innovation performance, and that the emotional experience of R&D personnel's freedom, openness, and trust in each other is more important for R&D efficiency enhancement than technological input.

In summary, a positive emotional experience has an important impact on collaborative innovation performance. Therefore, we propose the following hypothesis:

**Hypothesis 1c (H1c).** *Positive emotional experience will significantly enhance collaborative innovation performance.*

### 2.2. The Mediating Role of Collaborative Behavior between the Cognitive Characteristics of Innovation Teams and Collaborative Innovation Performance

Business cooperation is the operation of parallel firms in vertical and supply chain firms in horizontal to accomplish a unified goal or R&D innovation. Cooperation between enterprises is not only about the input of resources in the process of cooperation, but also about the active communication, participation, and coordination of innovation teams. Based on the length of cooperation, the breadth and depth of resource sharing, and the frequency of transactions, this study argues that the act of cooperation in a collaborative innovation network is a process in which firms establish connections of different natures based on the cognitive characteristics of innovation teams to achieve the goal of knowledge transfer and resource interchange.

In this collaborative process, firms facilitate the exchange and cooperation of innovation teams by establishing connections with other firms and sharing knowledge and resources. This collaborative behavior can be a long-term partnership or a short-term collaborative project aimed at jointly achieving innovation goals. By establishing collaborative relationships, firms are able to better exploit the cognitive characteristics of innovation teams in order to enable knowledge sharing and transfer, thereby driving innovation.

In conclusion, the act of corporate collaboration plays a crucial role in collaborative innovation networks. It is not only a reflection of the complementarity of resources and knowledge between enterprises, but also the result of the interaction between the cognitive characteristics of innovation teams and collaborative behavior. Through collaborative behavior, firms are able to work together to promote innovation and achieve a win–win situation.

### 2.2.1. Cognitive Characteristics and Collaborative Behavior of Innovative Teams

The social cognitive theory describes the response of individual behavior in organizations to psychological activity, where the cognitive decision-making process forms the underlying characteristics of behavior. In business, cognition is seen as an asset that must be assessed, developed, and managed because the key to behavioral motivation lies in the individual's cognition, a psychological dimension of activity that motivates perceptual behavior [38].

In the process of collaborative innovation, the effective use of explicit knowledge relies on the leverage of tacit knowledge. Firms with extensive experience of investing in R&D are, therefore, more likely to engage in external collaborative innovation. These firms increase their innovation capabilities and efficiency by establishing links with external partners and sharing knowledge and resources. The motivation and decisions for such collaborative innovation are based on firms' understanding and assessment of their perceptions of innovation [39].

When it comes to collaborative innovation, the cognitive differences between team members can also have an impact on collaborative behavior. Research has shown that cognitive differences among team members can promote innovative thinking and innovation [40]. Different cognitive backgrounds and experiences can bring about different perspectives and ideas, thus facilitating the occurrence and development of innovation. In addition, cognition is also closely related to team learning. Team learning is when team members learn and progress together by sharing knowledge, experience, and feedback. The sharing and understanding of cognition are the foundation of team learning, and they help to improve the team's problem-solving and innovation capabilities. In the collaborative innovation process, the cognition of individuals is also influenced by the external environment. The characteristics and climate of the innovation environment can stimulate innovative thinking and behavior in individuals. For example, companies with an organizational culture that encourages innovation and a management mechanism that supports innovation tend to stimulate employees' motivation and enthusiasm to innovate [41].

In conclusion, cognition plays an important role in collaborative innovation. Individual cognitive differences can promote innovative thinking and capabilities, while cognitive sharing and understanding among team members contribute to team learning and innovation capabilities. At the same time, the cognitive characteristics of the external environment can also have an impact on the innovation behavior of individuals and teams. Therefore, companies should pay attention to, and manage, the cognitive factors in the collaborative innovation process in order to promote the development and success of innovation.

Innovation experience refers to the knowledge, skills, and experience that an individual or team has accumulated from past innovation projects. First, rich innovation experience can enhance the trust and interaction among team members. This trust and interaction can promote cooperation and collaboration among the team members and improve the overall performance of the innovation team. Second, rich innovation experience can provide a common language and shared perceptions among the team members. In addition, specific to the collaborative innovation process, the effective use of explicit knowledge relies on the leverage of invisible knowledge, therefore, enterprises with extensive investment in the R&D experience are more inclined to participate in external collaborative innovation [42].

In summary, rich innovation experience has a significant impact on the collaborative behavior among innovation teams. By enhancing the trust and interaction among team members, as well as providing a common language and cognition, rich innovation experience can strengthen the collaborative behaviors of innovation teams, which in turn improves the performance and outcomes of innovation teams. Therefore, we propose the following hypothesis:

**Hypothesis 2a (H2a).** *A rich innovation experience will strengthen the collaborative behavior among innovation teams.*

A good internal innovation environment has an important impact on the collaborative behavior among innovation teams. Firstly, a good internal innovation environment creates a culture that supports collaboration and sharing. This cultural climate can stimulate mutual trust and mutual support among team members and promote cooperation and collaboration in innovation teams. Secondly, a good internal innovation environment can provide the necessary resources and support to facilitate innovation teamwork. Such resources and support can enhance complementarities and interactions among team members, thereby increasing the effectiveness of innovation teamwork [43].

In summary, by creating a cultural atmosphere that supports collaboration and sharing, as well as providing the necessary resources and support, a good internal innovation environment can strengthen the collaborative behaviors of innovation teams and further improve the performance and outcomes of innovation teams. Therefore, we propose the following hypothesis:

**Hypothesis 2b (H2b).** *A good internal innovation environment will strengthen the collaborative behavior among innovation teams.*

On the road to innovation, cooperation is an irreplaceable key element. Cooperative interaction and collaboration among team members can facilitate the breeding and development of ideas and bring the team to a higher level of innovation. However, cooperative behavior itself is also influenced by emotional factors. Positive emotions, such as positive moods, trust, and appreciation, may have a positive impact, further reinforcing collaborative behaviors among innovative teams. Inspired and motivated team members are more likely to share knowledge, exchange ideas, and support each other in the problem-solving process. Ref. [44] argued that the emotional experience of firms in the innovation process significantly affects their initiatives in areas such as information communication and determines whether firms are willing to provide adequate resource support for collaborative innovation activities. Therefore, we believe that positive emotions will play an important role in promoting the formation of a favorable collaborative atmosphere in innovation teams, thereby improving the innovation performance and overall team effectiveness. Therefore, the following hypothesis is proposed:

**Hypothesis 2c (H2c).** *Positive emotions will reinforce collaborative behavior among innovation teams.*

2.2.2. Collaborative Behavior and Collaborative Innovation Performance

Close communication between different innovation teams can form knowledge spillovers more effectively and thus reduce opportunity risks, which in turn can effectively enhance the performance of collaborative innovation [45]. In this regard, due to the self-protection awareness of enterprises, the quantity and quality of partners contacted by enterprises in the process of cooperation is related to the degree of access to resources, and the initiative of enterprises in the innovation network are key to improving the performance of collaborative innovation [46].

This close communication and collaboration helps to facilitate knowledge sharing and technology transfer, thus increasing the innovation capacity and efficiency of innovation teams. By sharing experiences, ideas, and resources, different innovation teams can learn from and build on each other, thereby accelerating the innovation process and reducing innovation risks. In addition, stable collaborative relationships provide continuous support and resources for companies to better respond to market changes and challenges.

In summary, close communication and collaboration between different innovation teams is crucial to the performance of collaborative innovation. By building stable collaborative relationships and proactively engaging in interactive innovation, firms can spread innovation costs, reduce innovation risks, and improve the performance of collaborative innovation. Such collaboration and communication can help to facilitate knowledge shar-

ing and technology transfer and improve the innovation capabilities and efficiency of innovation teams. Therefore, in the process of collaborative innovation, enterprises should pay attention to and actively promote close communication and cooperation between different teams.

From the above analysis, it can be concluded that the active participation of each innovation team in a collaborative innovation network will enhance the trust relationship among the members and thus efficiently enhance the output of the innovation. Therefore, the following hypothesis is proposed in this paper:

**Hypothesis 3 (H3).** *Deeply collaborative behavior will significantly enhance collaborative innovation performance.*

2.2.3. The Mediating Role of Cooperative Behavior

The above hypothesis suggests that there is a positive relationship between innovation team cognitive characteristics and cooperation behavior. Moreover, cooperation behavior also has a positive impact on collaborative innovation performance, therefore, it is easy to reason that innovation team cognitive characteristics may influence collaborative innovation performance through cooperation behavior. When firms do not have an innovation advantage, cooperation can compensate for the innovation disadvantage of individuals in the innovation network [47]. Clearly, universities, research institutes, and industry leaders with more innovation resources and the ability and willingness to collaborate with external parties are the best partners for firms to seek to improve their collaborative innovation performance [48].

Therefore, enterprises should actively establish collaborative relationships with these partners who have the innovation resources to enhance collaborative innovation performance [49]. Such cooperation can help enterprises to access more innovation resources and make up for their own shortcomings in the innovation network, thus improving the performance level of collaborative innovation. Cooperation with partners such as higher education institutions, research institutes, and industry leaders can help enterprises to share knowledge and technology, promote innovation capabilities, and thus improve the competitiveness of enterprises.

In summary, the cognitive characteristics of innovation teams have an impact on collaborative innovation performance through collaborative behavior. Collaborating with partners who have more innovation resources and are willing to cooperate is important for enterprises to improve their collaborative innovation performance. Enterprises should actively seek cooperation with partners such as universities, research institutes, and industry leaders in order to achieve an improved collaborative innovation performance.

From the above analysis it can be concluded that, in a diversified collaborative innovation network, the perceptions of innovation teams influence their collaborative behavior and the ability of firms to transfer, integrate, and utilize innovation resources, which in turn affects the collaborative innovation performance.

However, despite the important role that collaborative behaviors play in promoting innovation, we still lack a comprehensive understanding of the mechanisms that influence them. Therefore, there is a need to further explore the relationship between collaborative behavior in terms of innovation experience and co-innovation performance and to consider whether there is a mediating variable that can explain the link between the two. Based on this background, this hypothesis raises the possibility that collaborative behavior has a mediating role between innovation experience and co-innovation performance. By delving deeper into the research and validating this hypothesis, we can better understand the mechanisms by which collaborative behaviors influence co-innovation and provide organizations with suggestions for more effective innovation management strategies and practices. The hypothesis is as follows:

**Hypothesis 4a (H4a).** *Collaborative behavior has a mediating role in the relationship between innovation experience and collaborative innovation performance.*

The role of collaborative behavior in the innovation environment has been a matter of great interest. Collaborative innovation performance, on the other hand, is one of the most important indicators of a team's or organization's innovation capability. Based on this background, we propose a hypothesis that cooperative behavior may play a mediating role between the internal innovation environment and collaborative innovation performance. In other words, by promoting internal collaborative behaviors, a better innovation environment can be constructed, thus improving teams' co-innovation performance. This hypothesis is of great significance, as it provides new perspectives for understanding the relationship between various elements in the innovation process and useful references for exploring how to optimize the innovation environment to achieve a better co-innovation performance. Therefore, exploring the mediating role of the cooperative behaviors between the internal innovation environment and collaborative innovation performance is of great value when promoting the development of innovation management theories, as well as the enhancement of innovation capabilities in practice. The hypothesis is as follows:

**Hypothesis 4b (H4b).** *Collaborative behavior has a mediating role in the relationship between internal innovation environment and collaborative innovation performance.*

Emotional experience plays an important role in the process of collaborative innovation in organizations and teams. Emotional experience not only influences employees' positive mood and work motivation, but also shapes mutual trust and willingness to cooperate among team members. However, research suggests that emotional experience may not directly affect co-innovation performance, but rather play an indirect role through mediating variables. Based on this view, a hypothesis is proposed that collaborative behavior may have a mediating role between affective experience and co-innovation performance. This implies that, by promoting positive collaborative behaviors, not only can the affective experience of team members be enhanced, but also the co-innovation performance can be further improved. Such a hypothesis provides a new perspective that enables us to better understand the mechanisms by which affective experience affects co-innovation performance and provides organizations and teams with strategies and suggestions for improving affective experience to promote co-innovation. Therefore, exploring the mediating role of collaborative behavior in affective experience and co-innovation performance is of great significance in promoting the development of innovation capabilities in organizations and teams. The hypothesis is as follows:

**Hypothesis 4c (H4c).** *Collaborative behavior has a mediating role in the relationship between affective experience and collaborative innovation performance.*

*2.3. The Moderating Role of Team Innovation Efficacy in the Relationship between Collaborative Behavior and Collaborative Innovation Performance*

Self-efficacy is an important variable in the social cognitive theory, and with the rise of 'emotion research' in organizational behavior, explaining the mechanisms that trigger human behavior in terms of self-efficacy has become a new and popular topic [50]. In this regard, ref. [51] defines innovation efficacy as "an individual's internal beliefs about his or her ability to achieve creative outcomes," and suggests that the stronger the R&D member's confidence in innovation through teamwork, the more innovative ideas they will have. However, merely summing up individual self-innovation efficacy does not necessarily result in a shared sense of team innovation efficacy. Team innovation efficacy arises from the collective emergence and evolution of individual self-innovation efficacy at a team level, influenced by the development of social processes within the team. Their study revealed that creative self-efficacy serves as a partial mediator between the determinants and collective creative efficacy. Shin et al. [52] defines team innovation

efficacy as the shared, consistent perceptions of team members about their collective ability to innovate. The team's assessment of its innovation capabilities significantly influences the propensity to explore, and the innovation efficacy influences knowledge sharing within the innovation team and, thus, the innovation behavior of researchers and works by increasing the motivation of the team members and the effectiveness of such innovation [53].

A team's sense of innovation efficacy has a significant impact on the performance and the outcomes of innovative teams. When team members have a shared perception of the team's ability to innovate, they are more confident in their ability to produce innovations and are more willing to actively engage in exploratory innovation behaviors. A team's sense of innovation efficacy not only influences the sharing of knowledge within the team, but also has an impact on the innovative behavior of researchers. When team members believe that the team has the ability to innovate, they are more likely to share their knowledge and experience, promoting collaboration and innovation within the team. This positive behavior of knowledge sharing contributes to the team's effectiveness and ability to innovate. According to [54], teams that possess a high level of confidence in completing innovative tasks and achieving goals are more proactive in deviating from conventional behaviors during innovation activities. They exhibit a strong aversion to complacency and actively seek out challenges. Additionally, these teams are more inclined to embrace risk-taking behaviors and demonstrate a heightened level of perseverance when confronted with difficulties and obstacles.

In addition, a team's sense of innovation efficacy can also work by enhancing the members' behavioral motivation and innovation effectiveness. When team members believe that they and their team have the ability to innovate, they are more motivated to engage in innovative activities and are more motivated to overcome difficulties and challenges. This positive behavioral motivation and innovation effectiveness helps to drive the team's innovation process and achieve better innovation results.

Thus, the team's sense of innovation effectiveness plays an important role in their performance and outcomes. By sharing a consistent confidence and perception of innovation, team members are better able to collaborate and promote knowledge sharing and innovative behavior. At the same time, a team's sense of innovation efficacy can also motivate the members' behavior and innovation effectiveness, driving the team to achieve higher levels of innovation capability and performance.

Clearly, all of the above studies have highlighted the reinforcing relationship between team innovation efficacy and team behavior, especially when innovation teams have a high level of innovation efficacy, which strengthens the knowledge absorption and exploration capacity of the innovation team and subsequently increases the innovation output. Therefore, the following hypothesis is proposed in this paper:

**Hypothesis 5 (H5).** *Team innovation efficacy positively moderates collaborative behavior and collaborative innovation performance.*

In summary, this paper proposes the theoretical hypothesis model shown in Figure 1.

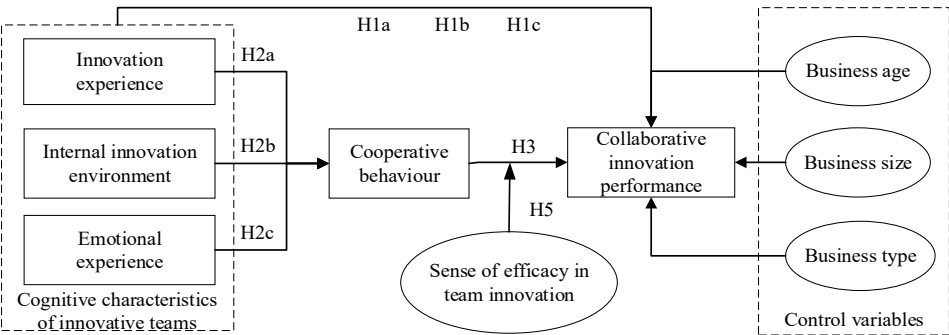

**Figure 1.** The theoretical hypothesis model.

## 3. Empirical Analysis

### 3.1. Sample Collection and Data Selection

Based on the five research hypotheses of this study, a survey team conducted a questionnaire survey to collect data from 35 teams consisting of 350 team members. An online questionnaire accounted for 60% and a traditional questionnaire accounted for 40%. Since the questionnaire was conducted in Chinese enterprises, Chinese questionnaires were selected. This research was mainly conducted on the domestic heavy usage of online survey platform "questionnaire star" (https://www.wjx.cn) through alumni network random distribution.

The sample teams were distributed across various industries, including, but not limited to, finance, manufacturing, and internet companies. Due to the impact of the pandemic, a combination of online and offline methods was used to distribute the questionnaires.

For the online survey, alumni networks were primarily utilized, while the offline survey relied on cooperation with partner companies involved in university–industry collaborative projects. It is worth noting that the majority of these partner companies were small- and medium-sized enterprises (SMEs). However, they demonstrated a fast iteration of knowledge and technology in university–industry collaborations, requiring ample resources for industrial technological upgrades. Additionally, these companies exhibited a high degree of alliance formation within the industry. They were more willing to engage in in-depth collaborative efforts for acquiring new knowledge and technologies. Therefore, they are considered representative of the industry.

A total of 350 questionnaires were distributed during the survey process, and 288 valid questionnaires were collected, resulting in an effective response rate of 82.3%. Considering the accuracy and validity of the data, the survey team obtained basic information about the participating teams before distributing the survey questionnaires. This information included the industry in which the teams were located, the size of the teams, and a basic profile of the team members, as shown in Table 1.

**Table 1.** Basic Information of Questionnaire Interviewees (n = 288).

| | Basic Information | Number | Percentage or Proportion |
|---|---|---|---|
| | He Nan Province | 124 | 43.06% |
| | Xin Jiang Province | 55 | 19.10% |
| Location of the Company | Jiang Su Province | 48 | 16.67% |
| | Shan Dong Province | 35 | 12.15% |
| | He Bei Province | 26 | 9.02% |
| Gender | Man | 194 | 67.36% |
| | Woman | 94 | 32.64% |
| | Master's degree and above | 51 | 17.71% |
| Level of Education | College degree (Associate's or Bachelor's) | 185 | 64.24% |
| | Below college degree | 52 | 18.05% |

According to Table 1, it can be observed that the companies were primarily located in Henan (43.06%), Xinjiang (19.10%), Jiangsu (16.67%), and so on. The basic information of those who completed the questionnaire was as follows: 194 (67.36%) were male and 94 (32.64%) were female; 52 (18.05%) were educated at college level or below; 185 (64.24%) were educated at college level or undergraduate level; and 51 (17.71%) had a master's degree or above. From the above research data, it can be seen that more men participated in this paper's research and a larger proportion of them had college level and above education.

### 3.2. Sources and Measures of Variables

This paper used the Likert 7-point scale to measure variables using a shared cognitive approach. The main criterion used was a 7-point scale of attitudes, on which questionnaire participants were asked to rate the team and their business, where: 1 = strongly disagree, 2 = disagree, 3 = rather disagree, 4 = fair, 5 = rather agree, 6 = agree, and 7 = strongly agree.

The internal innovation environment scale was based on the organizational innovation climate scale and used four items. The affective experience scale was based on the psychological feelings of team participation in collaborative innovation and asked respondents to rate three items. The measures of collaborative behavior were based on the definition of "strength of relationship" in the behavioral intentions scale and were measured with four items. The sense of team innovation efficacy was measured with four items, referring to the innovation efficacy scale. Based on the complexity of technological innovation, the output of innovation activities needed to meet the needs of specific targets. Therefore, this paper drew on the results of the above-mentioned scholars and used five items to measure the effectiveness of innovation (See Appendix A).

### 3.3. Exploratory Factor Analysis and Validation Factor Analysis

3.3.1. Exploratory Factor Analysis

(1)    Homogeneous ANOVA

Using Spss 21.0 to conduct homoscedasticity tests on 25 measures of 6 latent variables using maximum variance orthogonal rotations, the results of the 6 extracted common factors achieved a cumulative variance interpretation of 67.8% > 60% and the variance share of each extracted common factor was evenly distributed. This indicates that there was no homoscedastic bias in this paper and the six common factors were well represented, therefore, subsequent data analysis can be conducted.

(2)    Scale Reliability and Validity Test

The questions in this questionnaire were designed based on existing research and modified to meet the actual situation, with certain validity and reliability.

Firstly, Spss 21.0 was used to test the internal consistency of the scale. The results showed that the Cronbach's alpha coefficient for the innovation experience scale was 0.794, the Cronbach's alpha coefficient for the internal innovation environment was 0.843, the Cronbach's alpha coefficient for the emotional experience was 0.812, the Cronbach's alpha coefficient for the collaborative behavior scale was 0.756, and the Cronbach's alpha coefficient for the team innovation efficacy and collaborative innovation performance was 0.794. It is easy to see that the Cronbach's alpha coefficients of all of the scales were above 0.7, indicating that the reliability of the variables was high and good.

Secondly, the validity of the scales was tested by using the KMO test and Bartlett's sphere test. The results showed that the KMO values of the scales were all greater than 0.7, indicating that the questionnaire had good validity; moreover, the values of AVE were all greater than 0.5 and the values of combined reliability CR were all greater than 0.8, indicating that the scales had good convergent validity. The specific results are shown in Table 2.

**Table 2.** Reliability and Validity Test Scale.

|  | Title | Factor Load | Cronbach's $\alpha$ | KMO | AVE | CR |
|---|---|---|---|---|---|---|
| Innovation experience | JY1 | 0.826 | 0.794 | 0.721 | 0.624 | 0.869 |
|  | JY2 | 0.782 |  |  |  |  |
|  | JY3 | 0.748 |  |  |  |  |
|  | JY4 | 0.801 |  |  |  |  |
| Internal innovation environment | HJ1 | 0.892 | 0.843 | 0.778 | 0.684 | 0.896 |
|  | HJ2 | 0.881 |  |  |  |  |
|  | HJ3 | 0.791 |  |  |  |  |
|  | HJ4 | 0.734 |  |  |  |  |

**Table 2.** *Cont.*

|  | Title | Factor Load | Cronbach's $\alpha$ | KMO | AVE | CR |
|---|---|---|---|---|---|---|
| Emotional experience | QG1<br>QG2<br>QG3 | 0.858<br>0.872<br>0.821 | 0.812 | 0.704 | 0.724 | 0.887 |
| Cooperative behavior | HZ1<br>HZ2<br>HZ3<br>HZ4 | 0.797<br>0.754<br>0.804<br>0.691 | 0.756 | 0.719 | 0.582 | 0.847 |
| Sense of efficacy in team innovation | XN1<br>XN2<br>XN3<br>XN4 | 0.850<br>0.910<br>0.901<br>0.829 | 0.896 | 0.807 | 0.762 | 0.928 |
| Collaborative innovation performance | JX1<br>JX2<br>JX3<br>JX4<br>JX5 | 0.799<br>0.817<br>0.758<br>0.750<br>0.790 | 0.841 | 0.844 | 0.613 | 0.888 |

### 3.3.2. Validation Factor Analysis

Validated factor analysis (CFA) was conducted on the six variables using Amos 24.0 software, and four models ranging from single-factor to six-factor were developed in this paper to test the discriminant validity among the variables.

The data in Table 3 show that, among the six-factor models, GFI, IFI, TLI, and CFI were all greater than 0.9 and $\chi^2/df < 3$. The fit effect of the model, RMSEA < 0.08, met the test criteria and was significantly better than the other combinations, indicating that the discriminant validity of the variables was better and the fit of the model was higher.

**Table 3.** Indicators of goodness of fit of the measured model variables (n = 288).

| Variable | $\chi^2$ | df | $\chi^2/df$ | RMSEA | GFI | IFI | TLI | CFI |
|---|---|---|---|---|---|---|---|---|
| Standard |  |  | <3 | <0.08 | >0.9 |  |  |  |
| One-way test 1 + 2 + 3 + 4 + 5 + 6 | 1598.463 | 250 | 6.934 | 0.137 | 0.628 | 0.608 | 0.564 | 0.605 |
| Two-factor test 1 + 2 + 3 + 4, 5 + 6 | 1473.545 | 249 | 5.918 | 0.131 | 0.646 | 0.644 | 0.603 | 0.641 |
| Three-factor test 1 + 2 + 3, 4, 5 + 6 | 1034.048 | 247 | 4.186 | 0.105 | 0.742 | 0.771 | 0.742 | 0.77 |
| Four-factor test 1 + 2 + 3, 4, 5, 6 | 846.089 | 244 | 3.468 | 0.093 | 0.782 | 0.825 | 0.801 | 0.824 |
| Five-factor test 1, 2 + 3, 4, 5, 6 | 797.581 | 240 | 3.323 | 0.090 | 0.786 | 0.838 | 0.812 | 0.837 |
| Six-factor test 1, 2, 3, 4, 5, 6 | 369.2 | 235 | 1.571 | 0.045 | 0.905 | 0.961 | 0.954 | 0.961 |

### 3.3.3. Descriptive Statistics and Correlation Analysis

The results of the descriptive statistics and correlation analysis of the sample selection of each specific indicator using Spss21.0 are shown in Table 4. From Table 4, it can be seen that the team innovation experience was significantly and positively correlated with collaborative behavior and collaborative innovation performance (r = 0.503, $p < 0.01$; r = 0.516, $p < 0.01$); the internal innovation environment was significantly and positively correlated with collaborative behavior and collaborative innovation performance (r = 0.269, $p < 0.01$); the emotional experience was significantly and positively correlated with collaborative innovation performance (r = 0.327, $p < 0.01$; r = 0.586, $p < 0.01$); the collaborative behavior was significantly and positively correlated with collaborative innovation performance (r = 0.387, $p < 0.01$); and the team innovation efficacy was significantly and positively correlated with collaborative innovation performance (0.362, $p < 0.01$). The correlation coefficients between all of the main variables were significantly correlated.

**Table 4.** Correlation test (n = 288).

| | Average | Standard Deviation | Business Age | Business Size | Business Type | Innovative Experience | Internal Innovation Environment | Emotional Experience | Cooperative Behavior | Sense of Efficacy in Team Innovation | Collaborative Innovation Performance |
|---|---|---|---|---|---|---|---|---|---|---|---|
| Business age | 3.260 | 0.470 | 1 | | | | | | | | |
| Business size | 2.969 | 0.664 | 0.111 | 1 | | | | | | | |
| Business type | 2.448 | 1.119 | 0.192 ** | 0.277 ** | 1 | | | | | | |
| Innovative experience | 5.010 | 0.840 | 0.061 | −0.162 ** | −0.047 | (0.790) | | | | | |
| Internal innovation environment | 4.355 | 1.124 | 0.004 | −0.161 ** | −0.067 | 0.379 ** | (0.827) | | | | |
| Emotional experience | 5.117 | 0.875 | 0.093 | −0.112 | −0.015 | 0.639 ** | 0.362 ** | (0.851) | | | |
| Cooperative behavior | 4.797 | 0.944 | -0.052 | −0.155 ** | 0.006 | 0.503 ** | 0.327 ** | 0.486 ** | (0.763) | | |
| Sense of efficacy in team innovation | 5.013 | 0.966 | −0.059 | −0.100 | −0.010 | 0.445 ** | 0.421 ** | 0.393 ** | 0.520 ** | (0.873) | |
| Collaborative innovation performance | 5.506 | 0.760 | 0.098 | −0.038 | 0.038 | 0.516 ** | 0.269 ** | 0.586 ** | 0.387 ** | 0.362 ** | (0.783) |

Note: ** Significantly correlated at the 0.01 level (two-sided); square root of AVE in ().

In addition, the correlation coefficients between the variables were all smaller than the square root of the diagonal AVE, indicating that the results were less likely to be affected by multicollinearity and that the differential validity of the variables was good enough to be tested in the next step of the regression analysis. The specific results are shown in Table 4.

## 4. Research Findings and Discussion

Hierarchical linear regression analysis of the data using Spss 21.0 and the Process plug-in, with firm age, firm size, and firm type as the control variables, respectively, can verify the main effect of the innovation team cognitive characteristics on the collaborative innovation performance of firms, the mediating role of collaborative behavior in the innovation team cognitive characteristics and collaborative innovation performance. The results are shown in Table 5.

**Table 5.** Hierarchical linear regression analysis (n = 288).

| Variable | Collaborative Innovation Performance | | | | | | |
| --- | --- | --- | --- | --- | --- | --- | --- |
| | Model 1 | Model 2 | Model 3 | Model 4 | Model 5 | Model 6 | Model 7 |
| Business age | 0.098 | 0.055 | 0.071 | 0.0908 | 0.1093 ** | 0.0354 | 0.0494 |
| Business size | −0.058 | 0.028 | 0.043 | −0.0161 | 0.0243 | 0.0139 | 0.0292 |
| Business type | 0.035 | 0.044 | 0.031 | 0.0428 | 0.0189 | 0.0355 | 0.0268 |
| Innovative experience | | 0.519 *** | 0.429 *** | | | | |
| Internal innovation environment | | | | 0.269 *** | 0.162 ** | | |
| Emotional experience | | | | | | 0.586 *** | 0.515 *** |
| Cooperative behavior | | | 0.181 ** | | 0.343 *** | | 0.143 *** |
| Sense of efficacy in team innovation | | | | | | | |
| Cooperative behavior × Sense of efficacy in team innovation | | | | | | | |
| $R^2$ | 0.013 | 0.274 | 0.298 | 0.084 | 0.187 | 0.347 | 0.362 |
| F | 1.261 | 26.683 *** | 23.932 *** | 6.460 *** | 12.974 *** | 37.548 *** | 31.996 *** |

| Variable | Collaborative Innovation Performance | | | | Cooperative Behavior | | |
| --- | --- | --- | --- | --- | --- | --- | --- |
| | Model 8 | Model 9 | Model 10 | Model 11 | Model 12 | Model 13 | Model 14 |
| Business age | 0.116 * | −0.018 | 0.1845 * | −0.045 | −0.086 | −0.0539 | −0.0974 |
| Business size | 0.007 | −0.115 * | 0.0123 | −0.166 | −0.083 | −0.1175 ** | −0.1066 * |
| Business type | 0.011 | 0.046 | 0.0107 | 0.06 | 0.069 | 0.0695 | 0.0608 ** |
| Innovative experience | | | | | 0.498 *** | | |
| Internal innovation environment | | | | | | 0.313 *** | |
| Emotional experience | | | | | | | 0.484 *** |
| Cooperative behavior | 0.394 *** | | 0.240 *** | | | | |
| Sense of efficacy in team innovation | | 0.508 ** | 0.194 *** | | | | |
| Cooperative behavior × Sense of efficacy in team innovation | | | 0.095 ** | | | | |
| $R^2$ | 0.164 | 0.273 | 0.221 | 0.028 | 0.269 | 0.1234 | 0.2575 |
| F | 13.879 *** | 27.919 *** | 13.312 *** | 2.77 * | 25.973 *** | 9.9602 *** | 24.5349 *** |

Note: *** indicates $p < 0.001$, ** indicates $p < 0.01$, * indicates $p < 0.05$.

(1)　Main effects test

In order to test the influence of the cognitive characteristics of innovation teams on collaborative innovation performance, a regression model with collaborative innovation performance as the dependent variable was constructed. Model 1 is a base model with three control variables, including the age, size, and type of company in which the team of participants worked. Model 2, Model 4, and Model 6 were constructed by adding inno-

vation experience, internal innovation environment, and emotional experience to Model 1 to test the relationship between the team cognitive characteristics and the collaborative innovation performance. The results showed that the innovation experience, internal innovation environment, and emotional experience were all significantly and positively related to the collaborative innovation performance ($\beta$ = 0.519, $p < 0.001$; $\beta$ = 0.269, $p < 0.001$; and $\beta$ = 0.586, $p < 0.001$). This indicates that the collaborative innovation experience of the innovation team, the internal innovation environment of the company that they work for, and the positive emotion during innovation activities have significant positive effects on the collaborative innovation performance, among which, the collaborative innovation experience of the innovation team has the most significant effect on the collaborative innovation performance, followed by the emotional experience of the innovation team during innovation activities, while the internal innovation environment of the company that they work for has a relatively low degree of influence on the collaborative innovation performance. The degree of influence of the internal innovation environment of the innovation team on the collaborative innovation performance is relatively low. Therefore, hypotheses H1a, H1b, and H1c were verified.

(2) The intermediary role test

Firstly, to verify the influence of the team cognitive characteristics on cooperation behavior, a regression model with cooperation behavior as the dependent variable was constructed. Model 11 is the base model for the influence of the control variables on collaborative behavior, and Model 12, Model 13, and Model 14 were constructed by adding innovation experience, internal innovation environment, and emotional experience, respectively, to Model 11. The results show that the innovation experience, internal innovation environment, and emotional experience are significantly positively related to collaborative behavior ($\beta$ = 0.498, $p < 0.001$; $\beta$ = 0.313, $p < 0.001$; and $\beta$ = 0.484, $p < 0.001$), with the most significant effect of the innovation team's experience of collaborative innovation being on collaborative behavior, followed by the innovation team's emotional experience during innovation activities, and the relatively low effect of the internal innovation environment of the company in which the innovation team is located on collaborative behavior. Therefore, H2a, H2b, and H2c were verified.

Secondly, in order to verify the effect of collaborative behavior on collaborative innovation performance, Model 8 was constructed by adding collaborative behavior to Model 1. The results showed that collaborative behavior is significantly and positively related to collaborative innovation performance ($\beta$ = 0.394, $p < 0.001$). Therefore, hypothesis H3 was verified.

Finally, to verify the mediating role of collaborative behavior in the cognitive characteristics of innovation teams and collaborative innovation performance, Model 3, Model 5, and Model 7 were constructed in this paper using the Process plug-in, with method repeated sampling 5000 times, wherein, if the 95% confidence interval does not include the number 0, it indicates a mediating role. The results show that collaborative behavior plays a mediating role in the relationship between innovation experience, internal innovation environment, emotional experience, and collaborative innovation performance, with 95% upper and lower intervals not including 0. Therefore, hypotheses H4a, H4b, and H4c were verified. The specific regression coefficients and the Bootstrapping test results are shown in Table 6.

**Table 6.** Standardized Bootstrap intermediary role test.

|  |  | Effect | BootSE | BootLLCI | BootULCI |
|---|---|---|---|---|---|
| Innovative experience | Indirect effect | 0.082 | 0.028 | 0.030 | 0.138 |
|  | Direct effect | 0.388 | 0.057 | 0.275 | 0.497 |
|  | Total effect | 0.470 | 0.047 | 0.378 | 0.565 |
| Internal innovation environment | Indirect effect | 0.073 | 0.020 | 0.038 | 0.117 |
|  | Direct effect | 0.110 | 0.038 | 0.034 | 0.182 |
|  | Total effect | 0.182 | 0.037 | 0.110 | 0.254 |
| Emotional experience | Indirect effect | 0.060 | 0.024 | 0.018 | 0.111 |
|  | Direct effect | 0.448 | 0.050 | 0.346 | 0.542 |
|  | Total effect | 0.508 | 0.045 | 0.421 | 0.596 |

(3)    Regulation test

Model 9 was constructed on the basis of Model 1, and Model 10 was constructed by applying the Process plug-in on the basis of Model 8 and Model 9. The results show that the interaction term has a significant effect on collaborative innovation performance ($\beta$ = 0.095, $p < 0.01$), and using $\pm 1$ standard deviation as different levels of team innovation efficacy shows that team innovation efficacy positively moderates the effect of collaborative behavior on collaborative innovation performance. Therefore, hypothesis H5 was tested. The results of the moderating effect data at different levels are shown in Table 7 and the slope of the moderating effect at different levels is shown in Figure 2.

**Table 7.** The moderating effect of different levels of team innovation efficacy.

|  | Effect | BootSE | BootLLCI | BootULCI |
|---|---|---|---|---|
| Low team innovation efficacy | 0.148 | 0.057 | 0.035 | 0.261 |
| Medium team innovation efficacy | 0.240 | 0.051 | 0.141 | 0.340 |
| High team innovation efficacy | 0.332 | 0.065 | 0.205 | 0.459 |

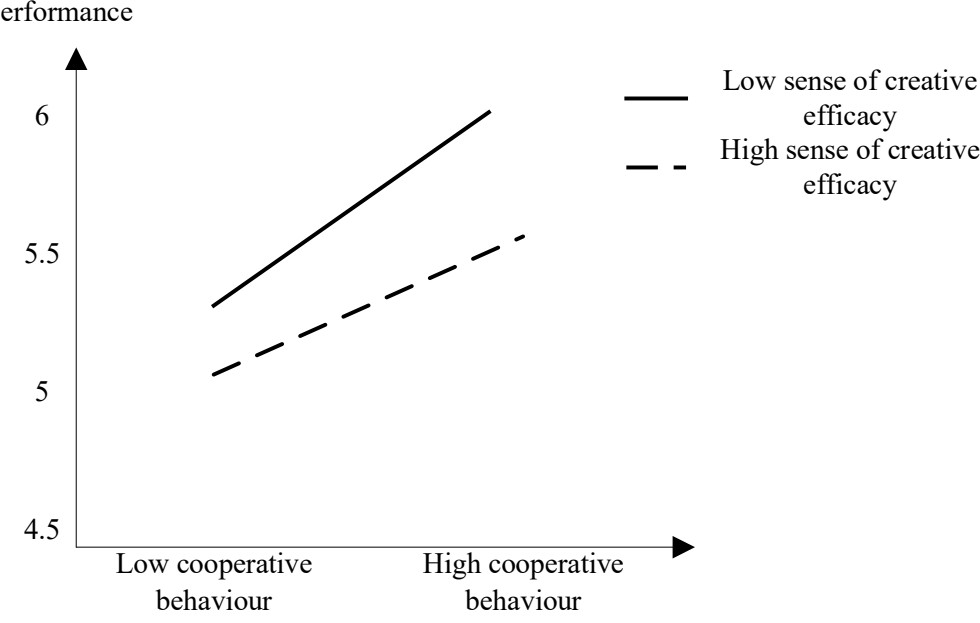

**Figure 2.** The moderating role of innovation efficacy on innovation perception.

### 5. Study Conclusions and Implications

*5.1. Study Conclusions*

This paper obtained the required survey data from the perspective of the social cognitive theory through joint university–enterprise innovation projects, alumni, and online questionnaire distribution and then empirically tested the mechanism of the influence of cognitive characteristics of innovation teams on collaborative innovation performance and the moderating effect of team innovation efficacy under the mediating role of cooperative behavior. The main findings of the study are as follows:

There is a positive relationship between the cognitive characteristics of innovation teams and collaborative innovation performance in the collaborative innovation process. This indicates that, when an innovation team within an enterprise innovates together with other innovation teams, and if its experience of innovation resources is realized to be transferred, it can reduce the cost of innovation while enhancing the efficiency of innovation synergy. At the same time, if the innovation team is supported or feels a strong innovation climate within the company that they are working for during the collaborative innovation process, the subjective level of consistency in perception will enhance the communication among the team members, generating a sense of trust and belonging among the individuals in the team and within the collaborative innovation network as a whole. In addition, the positive emotions of the innovation team can help to close the 'psychological distance' between members in specific collaborative innovation activities, thus contributing to the improvement of the collaborative innovation performance. The findings of this study echo the positive role of the internal innovation environment in collaborative innovation performance and extend the cognitive characteristics of innovation teams to the experiential and affective levels, to a certain extent deepening the role of the social cognitive theory in interpreting the impact mechanisms of collaborative innovation.

Collaborative behavior partially mediates the relationship between the cognitive characteristics of innovation teams and collaborative innovation performance. It suggests that firms with rich experience in collaborative innovation will enhance the investment of explicit innovation resources in order to maintain their own opportunity recognition ability. At the same time, a good internal innovation environment helps to reduce the worries of the innovation teams participating in innovation, thus facilitating the frequency of knowledge interaction between different teams. In addition, when innovation teams have positive emotional experiences, it means that their own efforts are recognized, which in turn can stimulate the innovation team's active innovation behavior. In this state, the company's vision of resource contribution and the innovation team's knowledge interaction initiatives are enhanced, which in turn leads to an improved collaborative innovation performance.

A team's sense of innovation efficacy positively moderates the relationship between collaborative behavior and collaborative innovation performance. When teams have a strong sense of innovation efficacy, psychological-level beliefs reinforce their ability to perform in collaborative innovation, which in turn demonstrates positive and innovative work attitudes and behaviors. At the same time, in addition to being more proactive in finding solutions to problems in uncertain innovation activities, innovation teams with a high level of innovation efficacy will proactively enhance the efficiency of the innovation team in resource utilization and knowledge transformation, thereby improving the collaborative innovation performance. This finding extends the existing literature and enriches the literature on the interaction of behavior and confidence in the social cognitive theory affecting collaborative innovation performance.

Finally, this study explores the key influencing factors of collaborative innovation activities from the perspective of the social cognitive theory by exploring the cognitive characteristics of innovation teams in corporate collaborative innovation networks, the mechanism of cooperation behavior on collaborative innovation performance, and the moderating role of team innovation efficacy. These findings help to further expand the scope of the application of the social cognitive theory in the field of collaborative innovation and pro-

vide theoretical support for the efficient mobilization of innovation initiatives of corporate innovation teams, so as to better promote the performance of collaborative innovation.

### 5.2. Research Insights and Prospects

In the current context of broad, deep, and diversified innovation, this paper offers a number of insights for enhancing the cognitive level of innovation teams, stimulating the motivation for innovation cooperation, and thus enhancing collaborative innovation performance, as follows:

Firstly, enriching the collaborative innovation experience of innovation teams and activating their sense of innovation efficacy requires that firms must help innovation teams to fully understand and use their individual capabilities to strengthen the trusting relationships between their internal members through organized collaborative innovation activities and increased support. At the same time, enterprises should develop incentive policies to encourage innovation in order to cultivate the confidence of the team in innovation and use tacit knowledge to promote the absorption and use of explicit knowledge, so as to reduce the management barriers to collaborative innovation and strengthen the initiative of the enterprise or innovation team.

Secondly, improving the internal innovation environment of enterprises requires that top managers must play a leading role in innovative thinking and use the spirit of innovation to lead knowledge learning in order to build a coherent goal for innovation teams. They should also develop an evaluation system for the internal innovation environment in order to improve the openness of innovation and focus on cultivating the independent learning ability of the internal members of the innovation team, which will then enhance the positive effect of the internal environment dynamics of the innovation team on external collaborative innovation.

Finally, optimizing the emotional experience of innovation teams requires that enterprises must build an information exchange platform, while breaking the barriers to the release of emotions and the expression of needs of innovation teams, and focus on strengthening communication, exchange, and cooperation between innovation teams in order to continuously improve the teams' collaborative innovation capability and achieve efficient innovation.

### 5.3. Theoretical Contribution

The theoretical contributions of this study are mainly reflected in the following aspects:

First, based on the social cognitive theory, this study explores the role of the cognitive characteristics of innovation teams in the collaborative innovation process. By considering the innovation team's experience of collaborative innovation, the firm's internal innovation environment, and affective experience as cognitive characteristics, we were able to gain a more comprehensive understanding of the team's behavior and performance in collaborative innovation. This helps to expand the application of the social cognitive theory in the field of collaborative innovation and provides a new perspective for theory construction.

Second, this study uses the positive behaviors of innovation teams during innovation activities that go beyond role prescriptions as a measure of collaborative behavior. Traditionally, collaborative behavior has often been confined to role prescriptions; however, innovation activities often require team members to act beyond their role prescriptions. By using positive behaviors beyond role prescriptions as a measure of collaborative behavior, this study was able to more accurately capture the performance of collaborative behavior in innovation teams during the collaborative innovation process.

Third, this study introduces the concept of team innovation efficacy and explores its role in the collaborative innovation process. Team innovation efficacy refers to team members' confidence and self-belief in the team's innovation capabilities and innovation performance. By investigating the mechanisms through which team innovation efficacy affects collaborative behavior and innovation performance, this study provides a new

perspective for understanding the mechanisms through which team innovation efficacy plays a role in collaborative innovation.

Overall, the theoretical contribution of this study is to extend the application of the social cognitive theory in the field of collaborative innovation and to introduce the concept of team innovation efficacy and explore its mechanism of action in the collaborative innovation process. This study provides a scientific reference basis for decision making to enhance the motivation of enterprises to participate in collaborative innovation and fills a gap in the research for related fields.

*5.4. Limitations*

Although this paper, based on the investigation of the influence of the cognitive characteristics of innovation teams on collaborative innovation performance, has verified part of the mediating role of cooperative behavior and the moderating role of team innovation efficacy, and has provided a scientific reference for the efficient implementation of innovation activities by innovation teams in enterprises, however, more dimensions of cognitive characteristics need to be explored in the future, and the interactions between the dimensions need to be considered in order to further improve the findings of this paper. Secondly, the endogeneity of the variables has not been further explored in this paper, and further in-depth research is needed in the future. And systematic hierarchical and typical multi-domain research should be conducted in the future to strengthen the applicability of the model.

**Author Contributions:** Software, W.Z.; Formal analysis, P.L.; Investigation, Y.J.; Data curation, X.W.; Writing—original draft, M.Z.; Writing—review & editing, M.X. All authors have read and agreed to the published version of the manuscript.

**Funding:** This research was funded by [General Project of Teacher Education Curriculum Reform in Henan Province: "Research on Reform of Talent Cultivation of Pre-school Education Majors Oriented on Vocational Competence in the Context of Double Reduction Policy"] grant number [2022-JSJYYB-117]; [General Project of Humanities and Social Sciences Research in Colleges and Universities in Henan Province: "Research on the Reform Path of Talent Cultivation of Preschool Education Majors Based on Vocational Competence under the Threshold of Double Reduction Policy"] grant number [2023-ZZJH-122]; [Research on Mechanism and Path of Innovation Management of Enterprises in Henan under the Strategy of Innovation Drive", Special Project of Cultural Research of Henan Xing Culture Project] grant number [2022XWH082]; [Research on Talent Cultivation of Industrial Engineering under the Perspective of Industry-University-Research Collaborative Education in Henan Province Education Science Planning Project] grant number [2022YB0014]; [Zhengzhou University Support Program Project for Young Talents and Enterprise Cooperative Innovation Team] grant number [2022117-32310385]; [2022 Academician Team Research Launch] grant number [13432340370]; [Zhengzhou University Graduate Students' Independent Innovation Project "From Great Manufacturing Province to Strong Smart Manufacturing Province—Research on Cultivating Digital Innovation Capability of Local Enterprises in Henan Province and Transformation and Application of Technological Achievements"] grant number [20230425]. And The APC was funded by [Peng Liu].

**Institutional Review Board Statement:** Not applicable.

**Informed Consent Statement:** Not applicable.

**Data Availability Statement:** The data presented in this study are not publicly available due to privacy restrictions.

**Conflicts of Interest:** The authors declare no conflict of interest.

## Appendix A

**Table A1.** The item measurement scale of the questionnaire.

| Variables | Items | Content |
|---|---|---|
| Innovation experience | JY1 | The team has participated in several collaborative innovation activities |
| | JY2 | The team has a great deal of expertise in innovation activities |
| | JY3 | The team has accumulated a lot of management skills during the innovation activities |
| | JY4 | The team has increased the sense of responsibility and mission in the innovation activities |
| Internal innovation environment | HJ1 | Team participation in collaborative innovation was strongly supported by the leadership |
| | HJ2 | The team and the rest of the organization encourage innovation |
| | HJ3 | The team and the rest of the organization are eager to learn and grow |
| | HJ4 | The team and other members of the organization often discuss innovation issues together |
| Emotional experience | QG1 | The team feels fair in the distribution of the benefits of collaborative innovation |
| | QG2 | The team is happy in collaborative innovation |
| | QG3 | The team's needs in collaborative innovation can be met |
| Cooperative behavior | HZ1 | Enterprises are very willing to support collaborative innovation activities with funds and resources |
| | HZ2 | The team is very willing to do their best in the cooperation process |
| | HZ3 | The team is very willing to communicate closely with other innovation teams |
| | HZ4 | The team is keen to promote the positive effects of innovation through collaboration |
| Sense of efficacy in team innovation | XN1 | The team is able to creatively complete innovative tasks |
| | XN2 | The team has the confidence that it can accomplish the expected goals |
| | XN3 | The team has the confidence to overcome challenges quickly and creatively |
| | XN4 | Teams are able to share resources in a collaborative innovation network and trust each other's capabilities |
| Collaborative innovation performance | JX1 | The collaborative innovation activities in which the team participated have led to significant improvements in existing products |
| | JX2 | The collaborative innovation activities that the team participated in created a lot of new technologies and applied for a lot of patents |
| | JX3 | The collaborative innovation activities that the team participated in resulted in the launch of many new products |
| | JX4 | Enterprises have gained more benefits than before they participated in collaborative innovation |
| | JX5 | Products created through collaborative innovation activities are well suited to market needs and even create new needs |

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
