# Peer review of "Cognitive Characteristics of an Innovation Team and Collaborative Innovation Performance: The Mediating Role of Cooperative Behavior and the Moderating Role of Team Innovation Efficacy"

_sustainability, doi:10.3390/su151410951_

Round 1

Reviewer 1 Report

Paper is standard and has potential but needs improvement in most of the areas.

 Introduction: The introduction needs to be more specific about the gaps and contributions. The context of your research in the field of innovation is too general. Articulate specific areas your study aims to address or gaps it seeks to fill. This will better position your research and its value in the existing body of literature. Also see innovation concept by Understanding the innovation concept of Maxamadumarovich, Obrenovic, Amonboyev.

 Literature and Hypotheses Development: The support for your hypotheses needs to be more extensive. Implement giving each hypothesis its own supporting paragraphs, which will not only provide a better foundation for your hypotheses but also make your paper more readable and organized. Especially for Hypothesis 2.2.1, additional support and elaboration would significantly strengthen your argument.

The review should be broadened and deepened by including more studies. With 39 studies referenced, there is room for including additional recent and relevant studies to build a stronger case. The following articles may be helpful:Can financial capability improve entrepreneurial performance? Evidence from rural China. Economic Research-Ekonomska Istraživanja, 36(1), 1631-1650.

Research on the energy poverty reduction effects of green finance in the context of economic policy uncertainty. Journal of Cleaner Production, 410, 137287.

Risk prediction in financial management of listed companies based on optimized BP neural network under digital economy. Neural Computing and Applications, 35(3), 2045-2058.

Monitoring and Early Warning of SMEs Shutdown Risk under the Impact of Global Pandemic Shock. Systems, 11(5), 260.

Does an imbalance in the population gender ratio affect FinTech innovation? Technological Forecasting and Social Change, 188, 122164.

From technology opportunities to ideas generation via cross-cutting patent analysis: Application of generative topographic mapping and link prediction. Technological Forecasting and Social Change, 192, 122565.

Management Power, R&D and Enterprise Performance: Moderating Effect Based on Management Competence. Journal of Chinese Human Resources Management, 12(1), 3-17.

Network power and innovation performance: Moderating effect based on knowledge base diversity and consistencyTaking new energy vehicle industry in Shanghai as an example. Journal of Chinese Human Resources Management, 12(2), 55-68.

Medical Device Product Innovation Choices in Asia: An Empirical Analysis Based on Product Space. Frontiers in public health, 10, 871575.

Emotion classification for short texts: an improved multi-label method. Humanities and Social Sciences Communications, 10(1), 306.

 Methodology: Your method section currently lacks the detail necessary for replication of the study. The scales used in the study should have sources or sample items provided for verification and reliability checks. The approach should be clarified in more detail to ensure the study's replicability and transparency.

 The article is original research paper rather than an essay. The distinction will clarify the nature and objectives of the research for  the readers, enhancing its academic integrity and value.

needs to be proofread and improved

Author Response

We appreciate the constructive comments provided by the three anonymous reviewers regarding the logic structure, terminology, and format of our manuscript. With your help, we have made a great effort to enhance the technical quality and presentation of the paper. The major modifications and additional analysis undertaken are as follows:

  • More specifically, the research gaps and contributions of the introduction are explained.
  • The review was further expanded and deepened to include additional studies.
  • The scale used in the study further provides source or sample items for validation and reliability checks.
  • Further supplement the theoretical contribution of the research.

Detailed responses to reviewers’ comments are presented in the following sections of this document. All the revised material in the revised version of the manuscript and our responses are marked in red.

Reviewer 2 Report

The paper is a reasonable effort, It provides valuable insights into the concepts of the innovation team, cooperative behavior, team innovation efficacy, and collaborative innovation performance. However, the authors need to make some amendments:

-> The practical research GAP is well explained, however, the theoretical gap is not well defined scientifically. Therefore, the authors are advised to revise and modify the introduction section with proper justification and arguments from the literature. In this regard, the authors are recommended to add theoretical/empirical gaps and contributions by recent studies.

-> Most of the literature is outdated. Perhaps authors can update the literature by adding the recent articles like mentioned below:

1.Subramaniam, M., Salleh, S. S. M. M., Suanda, J., Fareed, M., & Ahmad, A. (2023, April). Can technology innovation enrich employee performance? Evidence from Malaysian manufacturing industry. In AIP Conference Proceedings (Vol. 2544, No. 1). AIP Publishing.

-> Hypotheses development is weak. Authors can have separate discussions for each hypothesis like H1a, H1b, and H1c, the same goes for hypotheses H2a, H2b, H2c and H4a, H4b, and H4c.

-> There is not enough support for the moderating role of team innovation efficacy in the current study. Maybe authors need to cite those studies which have used team innovation efficacy as moderating variable with the collaborative innovation performance. Have previous studies shown an inconsistent relationship between collaborative behavior and collaborative innovation performance? Justify it with empirical evidence.

->Similarly more explanation is needed to test the mediating impact of collaborative behavior between cognitive characteristics of innovation teams and collaborative innovation performance.

->There is no information provided on population, sampling technique and sampling frame, and legitimacy of sample size.

-> Similarly no information is been given about respondents. Who are they? From whom data was collected? And why those respondents were suitable for the current study? Please justify it.

-> Present and document the quality of the methodology in a better way. To ensure the quality of the overall research process, the study must have rigor.

-> Perhaps authors can add a sample of questions from each variable that are being asked from the respondents in the methodology section in order to facilitate the reader to better understand what questions were being asked to form the findings of the study, if authors don't want to provide the complete instrument in the appendix section.

-> Additionally, either authors had adopted or adapted the instrument from previous scholars? No information is provided on from which studies instrument was adopted/adapted.

-> Please use the following article to justify the benchmark being met in Table 1 for Cronbach's alpha, composite reliability, and AVE values:

Abboh, U. A., Majid, A. H., Fareed, M., & Abdussalaam, I. I. (2022). High-performance work practices lecturers’ performance connection: Does working condition matter? Management in Education, 0(0). https://doi-org.eserv.uum.edu.my/10.1177/08920206211051468

-> Discussion is very weak. Needless to say, when it comes to discussing findings and contributions, we conform or differ from the work of previous scholars, in addition, to highlighting the unique contribution of our own work or how our work is different from the prior studies.

-> Please add a paragraph each on theoretical and practical implications as well as on future research directions.

The quality of English is fine. No problem is detected related to the English language or structures of the sentences.

Author Response

(The authors gave the same response as above.)

Reviewer 3 Report

Thank you so much for inviting me to review the manuscript sustainability-2461983 entitled “Cognitive characteristics of innovation team and collaborative innovation performance: The mediating role of cooperative behavior and the moderating role of team innovation efficacy.” I read it with great interest. I have the following comments

Introduction:

✓ In general, it is well written.
✓ Briefly discuss the theoretical contribution.

Literature review:

The literature section needs to be strengthened. The arguments are very brief and thin. Specifically, the mediator and moderator hypotheses required more arguments and upto-date literature.

Methods, Results, and Contribution:

The methodology section is very weak. I have many questions

How did you calculate the sample size?
Which sampling technique was used?
When the data collection was carried out?
Who was your targeted population?
Why you chose these three provinces?
Was the questionnaire in English or Chinese?
Which online platform was used?
What was the percentage division of online and traditional questionnaires?
The scales were adapted? From whom?
Add all scales in the appendix or Table 1 for clarity.

✓ The data were analyzed correctly and the results are interesting. However, the results should be compared with previous studies (wherever possible). In addition, also discuss how this study contributes to social cognitive theory.

✓ The managerial/practical implications are well presented but how this study contributes to the literature. Add the theoretical contribution of the study.

✓ Limitations and future research directions are fine.

Additional comments:
• Initially, write the word in full then use the abbreviation.
• Follow alphabetical order when more than one in-text citation.
• Follow journal guidelines.
• Proofreading is recommended.

Author Response

(The authors gave the same response as above.)

Round 2

Reviewer 1 Report

I can see that there have been improvements made to the paper. However certian sections were not fully addressed. I would like to see better use of the suggested literature to support the ideas behind your hypotheses and the introduction which consists out of innovation context and background.

The hypotheses must be reorganized with each being derived out of specific section, and not just all put together at the end of the section.

proofread further

Author Response

We appreciate the constructive comments provided by the three anonymous reviewers regarding the logic structure, terminology, and format of our manuscript. With your help, we have made a great effort to enhance the technical quality and presentation of the paper. The major modifications and additional analysis undertaken are as follows:

  • Reorganized the hypothesis section.
  • The research design, questions, hypotheses and methods were more clearly explained.

Detailed responses to reviewers’ comments are presented in the following sections of this document. All the revised material in the revised version of the manuscript and our responses are marked in red.

Reviewer 2 Report

Dear Authors,

Thank you very much for incorporating the comments in the updated manuscript.

Author Response

Thank you very much for reviewing my study and giving such a positive assessment. I feel very happy and encouraged by your compliments. Your professional comments and suggestions have been invaluable to me and have helped me to further improve my study. I am truly grateful for your time and effort and for your recognition of my work.I will continue to work hard to improve my research and look forward to receiving your valuable guidance again in the future.

Reviewer 3 Report

Thank you very much for inviting me to review the revised manuscript sustainability-2461983 entitled Cognitive characteristics of innovation team and collaborative innovation performance: The mediating role of cooperative behavior and the moderating role of team innovation efficacy.” I appreciate that author(s) have shown great commitment and dedication in revising the manuscript. I think they have answered most of my queries. However, I have still some comments/suggestions/concerns.

1.     How did you calculate the sample size? I would like to know using which method you concluded that 350 will be the appropriate number of questionnaires to distribute. What is the total number of the target population?

2.     Which sampling technique was used? Is it random or non-random and then which type?

3.  When the data collection was carried out? Your response is “2023-1.16~2023.4.27 but in the paper “a survey team conducted a questionnaire survey in December 2022 to collect data from 35 teams consisting of 350 teams members.” Please correct what is the exact time of your survey.

4.     Was the questionnaire in English or Chinese? Include the answer in the paper.

5.     Which online platform was used? Give details in the paper.

6.     What was the percentage division of online and traditional questionnaires? Include the details in the paper

For clarity, All these details should be in your paper.

Good luck.

Author Response

(The authors gave the same response as above.)
